# Dynamic Learning Rate for Deep Reinforcement Learning: A Bandit Approach

## Abstract

In deep Reinforcement Learning (RL), the learning rate critically influences both stability and performance, yet its optimal value shifts during training as the environment and policy evolve. Standard decay schedulers assume monotonic convergence and often misalign with these dynamics, leading to premature or delayed adjustments. We introduce LRRL, a meta-learning approach that dynamically selects the learning rate based on policy performance rather than training steps. LRRL adaptively favors rates that improve returns, remaining robust even when the candidate set includes values that individually cause divergence. Across Atari and MuJoCo benchmarks, LRRL achieves performance competitive with or superior to tuned baselines and standard schedulers. Our findings position LRRL as a practical solution for adapting to non-stationary objectives in deep RL.

## 1 Introduction

Reinforcement Learning (RL), when combined with function approximators such as Artificial Neural Networks (ANNs), has shown success in learning policies that outperform humans in complex games by leveraging extensive datasets (see, *e.g.*, Silver et al., 2016; Lample & Chaplot, 2017; Vinyals et al., 2019; Wurman et al., 2022). These achievements largely depend on training neural networks with gradient-based optimizers, where the choice of learning rate is critical to stability and convergence. An overly large learning rate can lead to divergence, while a small one can result in prohibitively slow learning (see Goodfellow et al., 2016; Blier et al., 2019; You et al., 2019). Despite its importance, selecting an appropriate learning rate in RL remains challenging due to two intertwined factors: the non-stationarity of the objective and the high cost of hyperparameter tuning.

Unlike supervised learning, where the loss function is typically stationary, RL agents learn from data generated by an evolving policy. As training progresses, both the distribution of visited states and the policy-induced rewards change, making the optimization landscape highly dynamic. Consequently, a learning rate that works well early in training may later become suboptimal. Prior work shows that even simple adjustments, such as reducing the learning rate in later stages, can improve performance in deep RL (Agarwal et al., 2022). However, designing an effective schedule is difficult because training steps do not necessarily reflect monotonic convergence toward a better policy.

Decay schedules, such as linear or exponential decay (Senior et al., 2013; You et al., 2019), reduce the learning rate over time based on predefined rules, assuming that the optimization problem becomes easier as training proceeds. This assumption often fails in RL, where progress can plateau or regress due to exploration challenges, sparse rewards, or environment stochasticity. Similarly, adaptive optimizers such as RMSProp (Tieleman & Hinton, 2012) and Adam (Kingma & Ba, 2015) adjust effective learning rates (see Lyle et al., 2024) based on past gradient statistics, which stabilizes updates but does not account for shifts in policy quality. Because these methods rely on step counts or gradient norms rather than environment feedback, they often make premature or delayed adjustments, limiting their ability to cope with RL's non-stationarity.

This motivates the question: **Can we adapt the learning rate dynamically based on the agent's performance, rather than relying on training progress or gradient-based heuristics?** A method satisfying this criterion should (i) leverage feedback that reflects policy quality rather than indirect signals like gradient norms, (ii) adapt to the different dynamics of early exploration and late-stage refinement, and (iii) remain general enough to integrate with a wide range of algorithms and optimizers.

We address this challenge by proposing **L**earning **R**ate for deep **R**einforcement **L**earning (LRRL), a meta-learning approach that casts learning rate adaptation as a dual objective: while the RL agent seeks to maximize cumulative returns, the bandit algorithm minimizes regret in selecting the most effective learning rates over time. LRRL formulates this as an adversarial multi-armed bandit (MAB) problem, where each arm corresponds to a candidate learning rate, and feedback is based on policy performance. By leveraging adversarial bandit algorithms with time-decay weighting, LRRL balances exploration of alternatives with exploitation of promising rates. Unlike fixed schedules or gradient-based meta-learning heuristics, LRRL is algorithm-agnostic, seamlessly integrates with optimizers such as Adam, RMSProp, and SGD, and enables dynamic adaptation across different phases of training.

We evaluate LRRL across a diverse set of scenarios, including discrete and continuous control tasks (Atari and MuJoCo), multiple deep RL algorithms (DQN, IQN, and PPO), and various optimizers. Our results show that LRRL achieves competitive or superior performance compared to fixed learning rates and standard decay strategies, while also demonstrating robustness to variations in configuration. Furthermore, to isolate RL-specific confounders, such as exploration and reward sparsity, we assess LRRL on stationary non-convex optimization problems using SGD, illustrating its ability to adaptively combine aggressive and conservative updates in a single optimization trajectory. Our main contributions are:

- We introduce LRRL, a novel approach that leverages a bandit-based controller to *select among a small set of learning rates on the fly* using policy performance as feedback. This directly targets RL's non-stationarity and avoids assuming monotonic convergence.

- We demonstrate *robustness to poorly performing candidates*: LRRL remains stable and competitive even when the candidate set contains rates that individually fail, by adapting toward effective rates during training.

- We evaluate LRRL using a set of learning algorithms on Atari and MuJoCo tasks, showing performance *competitive with or superior* to tuned fixed-rate training and decay schedulers, alongside ablations of key design parameters.

- Beyond RL, we illustrate how combining multiple rates within a single run can be beneficial in stationary non-convex optimization with SGD, underscoring LRRL's generality.

## 2 PRELIMINARIES

This section introduces the RL and MAB frameworks and defines the supporting notation.

### 2.1 DEEP REINFORCEMENT LEARNING

An RL task is defined by a Markov Decision Process (MDP), that is, by a tuple $(\mathcal{S}, \mathcal{A}, P, R, \gamma, T)$, where $\mathcal{S}$ denotes the state space, $\mathcal{A}$ the set of possible actions, $P : \mathcal{S} \times \mathcal{A} \times \mathcal{S} \rightarrow [0, 1]$ the transition probability, $R : \mathcal{S} \times \mathcal{A} \rightarrow \mathbb{R}$ the reward function, $\gamma \in [0, 1]$ the discount factor, and $T$ the horizon length in episodic settings (see *e.g.*, Sutton & Barto, 2018 for details). In RL, starting from an initial state $s_0$, a learner called *agent* interacts with the environment by picking, at time $t$, an action $a_t$ depending on the current state $s_t$. In return, it receives a reward $r_t = R(s_t, a_t)$, reaching a new state $s_{t+1}$ according to the transition probability $P(\cdot|s_t, a_t)$. The agent's objective is to learn a policy $\pi(\cdot|s)$ that maps a state $s$ to a probability distribution over actions in $\mathcal{A}$, aiming to maximize expected returns $Q^\pi(s, a) = \mathbb{E}_\pi \left[ \sum_{t=0}^{T} \gamma^t r_t \mid s_0 = s, a_0 = a \right]$.

To learn how to perform a task, value function-based algorithms coupled with ANNs (Mnih et al., 2013; 2015) approximate the *quality* of a given state-action pair $Q(s, a)$ using parameters $\theta$ to derive a policy $\pi_\theta(\cdot|s) = 1_{a^*(s)}(\cdot)$ where $1_{a^*(s)}$ denotes the uniform distribution over $a^*(s)$ with $a^*(s) = \arg\max_{a \in \mathcal{A}} Q_\theta(s, a)$. By storing transitions $(s, a, r, s') = (s_t, a_t, r_t, s_{t+1})$ in replay memory $\mathcal{D}$, the objective is to minimize the loss function defined by:

$$\mathcal{J}(\theta) = \mathbb{E}_{(s,a,r,s')\sim\mathcal{D}} \left[ r + \gamma \max_{a' \in \mathcal{A}} Q_{\theta^-}(s', a') - Q_\theta(s, a) \right]^2, \tag{1}$$

where $\theta^-$ are the parameters of the target network used to calculate the target of the learning network $y = r + \gamma \max_{a' \in \mathcal{A}} Q_{\theta^-}(s', a')$. The parameters $\theta^-$ are periodically updated by copying the parameters $\theta$, leveraging stability during the learning process by fixing the target $y$. The minimization,

and hence the update of the parameters $\theta$, is done according to the optimizer's routine. A simple possibility is to use SGD using mini-batch approximations of the loss gradient:

$$\theta_{n+1} \leftarrow \theta_n - \eta \nabla_\theta \mathcal{J}(\theta_n) \, , \text{where}$$

$$\nabla_\theta \mathcal{J}(\theta_n) \approx \frac{1}{|\mathcal{B}|} \sum_{(s,a,r,s') \in \mathcal{B}} 2\big(Q_\theta(s,a) - y\big) \nabla_\theta Q_\theta(s,a), \tag{2}$$

with $\mathcal{B}$ being a mini-batch of transitions sampled from $\mathcal{D}$ and $\eta$ is a single scalar value called *learning rate*. Unlike supervised learning, where the loss function $\mathcal{J}(\theta)$ is typically stationary, RL presents a fundamentally different challenge: the policy is continuously evolving, leading to shifting distributions of states, actions, and rewards over time. This continuous evolution introduces instability during learning, which deep RL mitigates by employing a large replay memory and calculating the target using a *frozen* network with parameters $\theta^-$. However, stability also depends on how the parameters $\theta$ change during each update. This work aims to adapt to these changes by dynamically selecting the learning rate $\eta$ in the training steps.

## 2.2 MULTI-ARMED BANDITS

MAB provide an elegant framework for making sequential decisions under uncertainty (see for instance Lattimore & Szepesvári, 2020). MAB can be viewed as a special case of RL with a single state, where at each round $n$, the agent selects an arm $k_n \in \{1, \ldots, K\}$ from a set of $K$ arms and receives a feedback (reward) $f_n(k_n) \in \mathbb{R}$. Like RL, MAB algorithms must balance the trade-off between exploring arms that have been tried less frequently and exploiting arms that have yielded higher rewards up to time $n$.

To account for the non-stationarity of RL rewards, we will consider in this work the MAB setting of adversarial bandits (Auer et al., 2002). In this setting, at each round $n$, the agent selects an arm $k_n$ according to some distribution $p_n$ while the environment (the *adversary*) arbitrarily (*e.g.*, without stationary constraints) determines the rewards $f_n(k)$ for all arms $k \in \mathcal{K}$. MAB algorithms are designed to minimize the *pseudo-regret* $G_N$ after $N$ rounds defined by:

$$G_N = \mathbb{E}\left[\sum_{n=1}^{N} \max_{k \in \{1, \ldots, K\}} f_n(k) - \sum_{n=1}^{N} f_n(k_n)\right],$$

where the randomness of the expectation depends on the MAB algorithm and on the adversarial environment, $\sum_{n=1}^{N} \max_{k \in \{1, \ldots, K\}} f_n(k)$ represents the accumulated reward of the potentially changing single best arm, and $\sum_{n=1}^{N} f_n(k_n)$ is the accumulated reward obtained by the algorithm. A significant component in the adversarial setting is to ensure that each arm $k$ has a non-zero probability $p_n(k) > 0$ of being selected at each round $n$. This guarantees exploration, which is essential for the algorithm's robustness to environment changes.

## 3 DYNAMIC LEARNING RATE FOR DEEP RL

In this section, we introduce LRRL, a meta-learning approach for dynamically adapting the learning rate based on policy performance. LRRL addresses the challenge of non-stationarity in RL by framing learning rate selection as a multi-armed bandit (MAB) problem. While the RL policy aims to maximize cumulative return, the bandit algorithm minimizes regret in identifying the most effective learning rates over time. Each candidate learning rate corresponds to an arm, and the bandit uses policy performance as feedback. We adopt an adversarial bandit algorithm with time-decay weighting to accommodate the evolving reward distribution as the agent interacts with the environment.

### 3.1 SELECTING THE LEARNING RATE DYNAMICALLY

Our problem can be framed as selecting a learning rate $\eta$ for policy updates —specifically, when updating the parameters $\theta$ after $\lambda$ interactions with the environment. Before training, the user defines a set $\mathcal{K} = \{\eta_1, \ldots, \eta_K\}$ of $K$ learning rates. Then, during training, a MAB algorithm selects, at every round $n$ —that is, at every $\kappa$ interactions with the environment— an arm $k_n \in \{1, \ldots, K\}$ according to a probability distribution $p_n$ defined based on previous rewards, as explained in the next

---

**Algorithm 1** dynamic Learning Rate for deep Reinforcement Learning (LRRL)

---

**Initialize:** network parameters $\theta$, target parameters $\theta^-$, arm probabilities $p_0 \leftarrow (\frac{1}{K}, \ldots, \frac{1}{K})$, MAB round $n \leftarrow 0$, cumulative reward $R \leftarrow 0$, counter $C \leftarrow 0$

**for** episode $m = 1, 2, 3, \ldots, M$ **do**
   **for** timestep $t = 1, 2, 3, \ldots, T$ **do**
      Choose action $a_t$ following $\pi_\theta(\cdot|s)$ with probability $1 - \epsilon$     ($\epsilon$-greedy strategy)
      Play action $a_t$ and observe reward $r_t$
      Add $r_t$ to cumulative reward $R \leftarrow R + r_t$
      Increase interactions counter $C \leftarrow C + 1$
      **if** $C \mod \lambda \equiv 0$ **then**
         **if** $C \geq \kappa$ **then**
            Compute average $f_n \leftarrow \frac{R}{C}$
            Increase MAB round $n \leftarrow n + 1$
            Compute weights $w_n$ and arm probabilities $p_n$ using Equations (3, 4)
            Sample arm $k_n$ with distribution $p_n$
            Reset $R \leftarrow 0, C \leftarrow 0$
         **end if**
         Update network parameter $\theta$ using the optimizer update rule with learning rate $\eta_{k_n}$
         Every $\tau$ steps update the target network $\theta^- \leftarrow \theta$
      **end if**
   **end for**
**end for**

---

section. The parameters $\theta$ are then updated using the sampled learning rate $\eta_{k_n}$. The steps involved in this meta-learning approach are summarized in Algorithm 1.

Note that the same algorithm might be used with learning rates schedulers, that is with $\mathcal{K} = \{\eta_1, \ldots, \eta_K\}$ where $\eta_k : \mathbb{N} \to \mathbb{R}_+$ is a predefined function, usually converging towards 0 at infinity. If so, the learning rate used at round $n$ of the optimization is $\eta_{k_n}(n)$.

### 3.2 Updating the Probability Distribution

Since the agent's performance—and thus the cumulative reward—changes over time, the MAB algorithm naturally receives non-stationary feedback. To take this non-stationary nature of the learning into account, we employ a modified version of the Exponential-weight algorithm for Exploration and Exploitation (Exp3, see Auer et al., 2002 for an introduction). At round $n$, Exp3 chooses the next arm (and its associated learning rate) according to the arm probability distribution $p_n$ which is based on weights $(w_n(k))_{1 \leq k \leq K}$ updated recursively. Those weights incorporate a time-decay factor $\delta \in (0, 1]$ that increases the importance of recent feedback, allowing the algorithm to respond more quickly to changes of the best action and to improvements in policy performance. Specifically, after picking arm $k_n$ at round $n$, the RL agent interacts $C$ times with the environment and the MAB algorithm receives a feedback $f_n$ corresponding to the average reward of those $C$ interactions. Based on Moskovitz et al. (2021), this feedback is then used to compute the *improvement in performance*, denoted by $f'_n$, obtained by subtracting the average of the past $j$ bandit feedbacks from the most recent one $f_n$:

$$f'_n = f_n - \frac{1}{j} \sum_{i=0}^{j-1} f_{n-i} \, .$$

The improvement in performance allows the computation of the next weights $w_{n+1}$ as follows:

$$\forall k \in \{1, \ldots, K\}, \qquad w_{n+1}(k) = \begin{cases} \delta \, w_n(k) + \alpha \, \frac{f'_n}{e^{w_n(k)}} & \text{if } k = k_n \\ \delta \, w_n(k) & \text{otherwise} \, , \end{cases} \tag{3}$$

where initially $w_1 = (0, \ldots, 0)$ and $\alpha > 0$ is a step-size parameter. The distribution $p_{n+1}$, used to draw the next arm $k_{n+1}$, is defined by:

$$\forall k \in \{1, \ldots, K\}, \quad p_{n+1}(k) = \frac{e^{w_{n+1}(k)}}{\sum_{k'=1}^{K} e^{w_{n+1}(k')}} \, . \tag{4}$$

This update rule ensures that as the policy $\pi_\theta$ improves the cumulative reward, the MAB algorithm continues to favor learning rates that are most beneficial under the current policy performance, thereby effectively handling the non-stationarity inherent in the learning process.

## 4 EXPERIMENTS

In this section, we evaluate LRRL across both discrete and continuous control tasks, using multiple RL algorithms and optimizers. Our study includes Atari games (discrete control, DQN and IQN agents) with baselines provided by the Dopamine framework (Castro et al., 2018), and MuJoCo tasks (continuous control, PPO agents) with baselines implemented in TorchRL (Bou et al., 2024). For Atari, we report the mean and one-half standard deviation of returns over 5 independent seeds, while for MuJoCo we use 10 seeds per algorithm. Across all tasks, we compare LRRL against fixed learning rates and schedulers, combined with widely used optimizers such as Adam and RMSProp.

As testing the full Atari benchmark is computationally prohibitive, we select a representative subset of games guided by their baseline learning curves, capturing a range of properties such as reward sparsity, stochasticity, and exploration difficulty. To further disentangle RL-specific confounders, such as environment stochasticity and exploration dynamics, we also evaluate LRRL with SGD on stationary non-convex optimization problems, providing a controlled setting where performance can be attributed solely to learning rate adaptation. Additional details on evaluation metrics and hyperparameters are provided in Appendices D and E.

### 4.1 LRRL VS. FIXED LEARNING RATES: ROBUSTNESS ACROSS CANDIDATE SETS

In our first experiment, we evaluate whether LRRL can remain competitive with tuned baselines when learning rates are drawn from different candidate sets. We consider five possible learning rates,

$$\mathcal{K}(5) = \left\{ 1.5625 \times 10^{-5}, 3.125 \times 10^{-5}, 6.25 \times 10^{-5}, 1.25 \times 10^{-4}, 2.5 \times 10^{-4} \right\}.$$

and construct multiple LRRL variants based on different subsets: the full set $\mathcal{K}(5)$, the three smallest rates $\mathcal{K}_{lowest}(3)$, the three middle rates $\mathcal{K}_{middle}(3)$, the three largest rates $\mathcal{K}_{highest}(3)$, and a sparse mix of low, medium, and high values $\mathcal{K}_{sparse}(3)$. All experiments use the Adam optimizer.

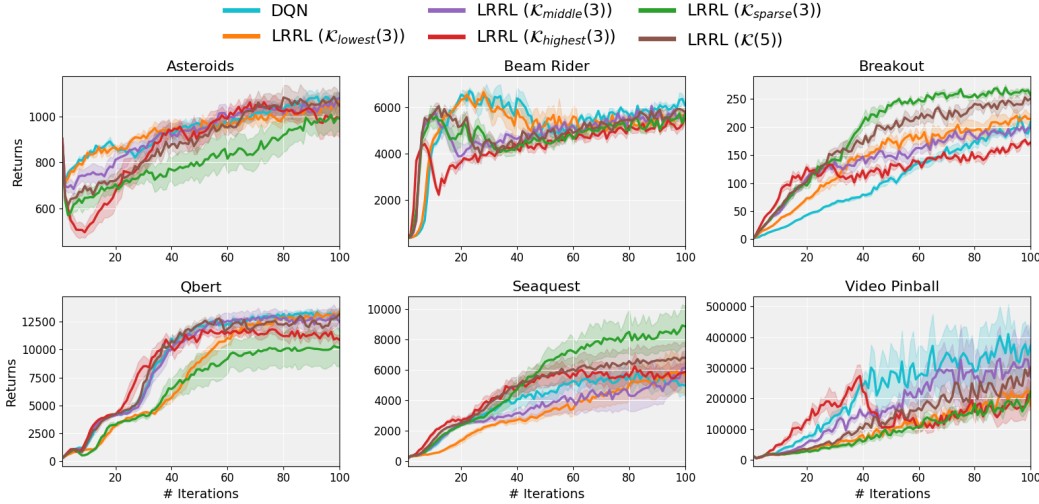

Figure 1: LRRL and standard DQN comparison: 5 variants of LRRL with different learning rate sets are tested against the DQN algorithm reaching best performance among the possible learning rates (provided in Appendix 7). The mean and one-half standard deviations over 5 runs are represented.

It is important to note that the performance of LRRL naturally depends on the composition of the candidate set. If most of the available learning rates perform poorly in isolation, then LRRL's performance will also be limited, as it can only adapt within that set. This does not indicate a failure of the method, but rather reflects the quality of the available candidates. Conversely, when the set contains both effective and ineffective rates, LRRL consistently adapts toward the better-performing ones, maintaining stability even when poor candidates are present.

In this experiment, we compare LRRL against the best single fixed rate among the candidates. This baseline represents an oracle that assumes prior knowledge of the best-performing learning rate, requiring multiple training runs. LRRL, by contrast, adapts within a single run.

To better understand how this adaptation occurs, we analyze the sequence of learning rates selected during training. Figure 2 visualizes the arms pulled by LRRL $\mathcal{K}(5)$ and the corresponding returns from a single run. In most environments, LRRL initially favors higher learning rates to accelerate early progress, before gradually shifting toward smaller ones as performance stabilizes. This behavior resembles a decay schedule, but crucially, it emerges from policy performance feedback rather than being predefined. The ability to combine aggressive and conservative updates within a single trajectory illustrates how LRRL dynamically adapts to non-stationary learning dynamics.

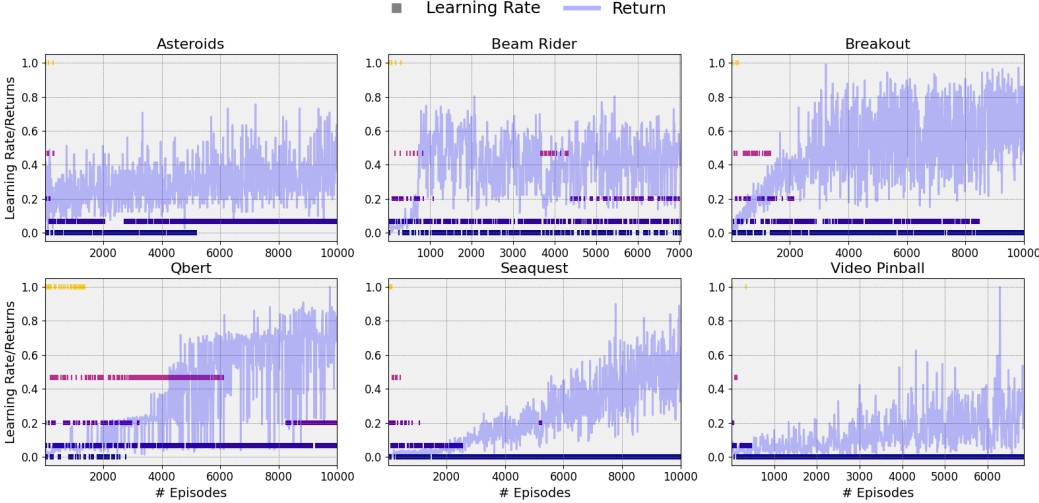

Figure 2: Systematic sampling of normalized learning rates and returns over the training steps using LRRL $\mathcal{K}(5)$ with Adam optimizer, through a single run. For each episode, we show the selected learning rate using different colors. Lower rates are increasingly selected over time.

## 4.2 LRRL with Learning Rate Schedulers

We next investigate whether LRRL can improve upon fixed learning rate schedulers by adapting among them during training. Specifically, we consider exponential decay schedules of the form $\eta(n) = \eta_0 \times e^{-dn}$, where $\eta_0 = 6.25 \times 10^{-5}$ is a fixed initial value, $d$ is the exponential decay rate, and $n$ is the number of policy updates (*i.e.*, MAB rounds). We define a candidate set $\mathcal{K}_s$ of three schedulers with $d \in 1, 2, 3 \times 10^{-7}$, and compare LRRL against each scheduler used individually, with Adam as the optimizer.

Figure 3 shows that LRRL achieves clear gains in two out of six tasks and remains competitive in the others. Importantly, LRRL consistently improves jumpstart performance in most of tasks. This highlights a key advantage over fixed schedulers: whereas decay schedules impose a monotonic reduction in learning rate, LRRL adapts the choice dynamically based on policy performance. For reference, the dashed black line indicates the performance of Adam with a constant learning rate, which is generally slightly weaker than the best-performing decayed scheduler, in line with previous findings (Andrychowicz et al., 2021).

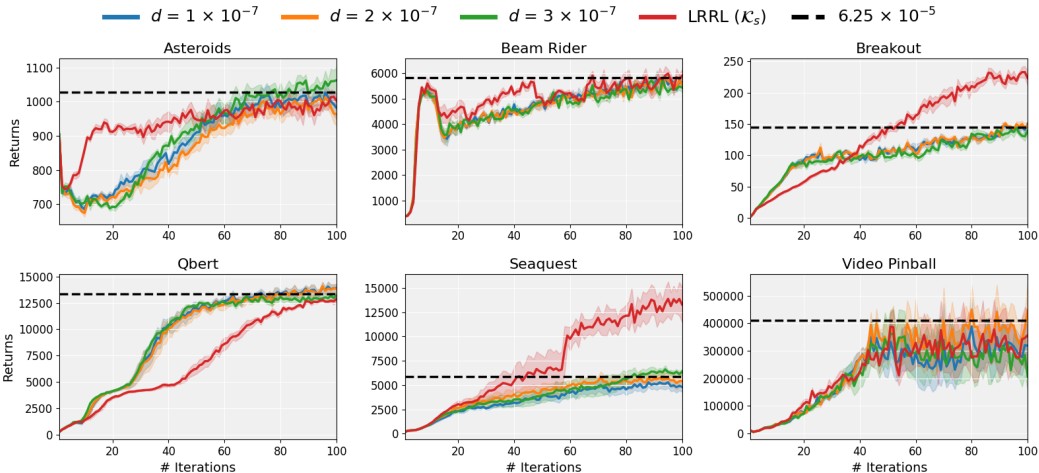

Figure 3: LRRL vs 3 individual schedulers ($d = 1, 2, 3$). The dashed black line represents the max average return achieved by Adam with a constant learning rate set to $6.25 \times 10^{-5}$.

**Effect of the Update Window $\kappa$.** In the experiments so far, we have fixed the update window $\kappa$ to 1, meaning LRRL observes a complete episode before updating the learning rate. However, in highly stochastic environments, updating after every episode may provide noisy feedback, making it harder for LRRL to identify the most effective learning rate. Increasing $\kappa$ allows more interactions before each update, trading off reactivity for stability. Two factors should guide this choice:

- **Network update rate:** If learning rates are switched more frequently than model updates occur, the feedback to the bandit become inaccurate.

- **Environment feedback:** In tasks where outcomes are only revealed after long action sequences, observing more steps before updating helps capture agent performance more reliably.

We evaluate LRRL with different values of $\kappa$ when combined with schedulers. As shown in Figure 4, the effect of $\kappa$ is task-dependent: some games benefit from more reactive updates, while others gain from longer observation windows. Overall, $\kappa = 1$ remains a robust choice for Atari games, though higher values can yield slight improvements in certain tasks.

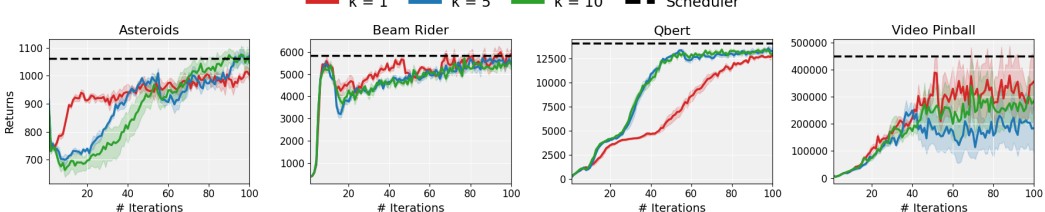

Figure 4: Results of the ablation study evaluating the impact of varying $\kappa$ with LRRL using schedulers. The dashed black line represents the max average return achieved by the best performing scheduler.

## 4.3 CONTINUOUS ACTION SPACES

In the following, we evaluate LRRL on MuJoCo tasks using Proximal Policy Optimization (PPO) (Schulman et al., 2017) implemented in TorchRL (Bou et al., 2024), which by default applies a linear decay schedule to both actor and critic learning rates.

We compare LRRL against two alternative strategies: *AdamRel* (Ellis et al., 2024) and *Cyclical Learning Rates (CLR)* (see details in B.5), the latter of which has shown promising results in RL despite higher instability (Lyle et al., 2025). To isolate the effect of the adaptation mechanism (rather than LR range), we adopt the following implementation choices for this comparison: (i) CLR cycles deterministically over the *same candidate learning rate set* used by LRRL; (ii) AdamRel uses the standard fixed learning rate (no decay) in this experiment[1]; and (iii) actor and critic share the same learning rate across methods. We report means and one-half standard deviations over 10 seeds, evaluating policies after each batch for a total of 1M frames.

As shown in Figure 5, LRRL achieves higher returns on two of five MuJoCo tasks. Unlike CLR, which cycles through rates without a performance-based criterion, LRRL adapts its selection from the *same* pool using policy feedback, leading to stronger or comparable outcomes in most of the tasks. Compared to AdamRel, LRRL does not rely on optimizer-specific mechanics (momentum resets) and is applicable across different optimizers while still adapting to non-stationary learning dynamics.

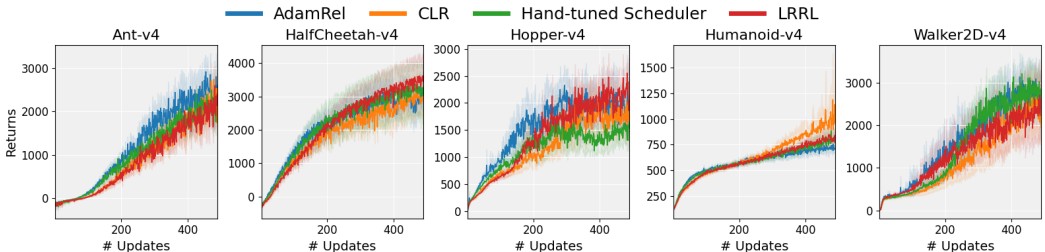

Figure 5: Comparison of LRRL, AdamRel, and CLR on MuJoCo tasks using PPO. LRRL achieves higher average returns in two tasks. Lines represent the mean and shaded regions show one-half standard deviations over 10 seeds, with training run for 1M frames.

## 4.4 SGD AND STATIONARY NON-CONVEX LOSS LANDSCAPES

Although RL tasks are a natural showcase for LRRL under non-stationarity, they involve many confounding factors, such as the adaptive routine of the optimizer, stochastic environment dynamics, and reward sparsity. To disentangle these effects, we evaluate LRRL in a simpler, stationary setting: minimizing six standard non-convex loss landscapes using stochastic gradient descent (SGD). Unlike Adam or RMSProp, SGD employs a single scalar learning rate without momentum or adaptivity, making it an ideal testbed to isolate the impact of dynamic learning rate selection.

**Experimental setup.** Each minimization works for a fixed budget of steps from the same initialization. LRRL is applied on top of SGD using the stochastic bandit Minimax Optimal Strategy in the Stochastic case (MOSS; see Appendix B.2), with arms corresponding to candidate learning rates. At each gradient step $n$, the bandit receives the feedback

$$f_n = \frac{1}{\mathcal{J}(w_1^{(n)}, w_2^{(n)}) + \xi}, \quad \xi > 0,$$

where $\mathcal{J}$ denotes the loss and $\xi$ prevents division by zero. The bandit updates after every gradient step. Figure 6 illustrates the optimization trajectories, with final iterates marked by stars. Two consistent behaviors emerge:

• **Adapting across multiple rates.** On landscapes such as Beale, Bohachevsky, and Griewank, LRRL flexibly alternates between larger and smaller learning rates depending on the stage of optimization. By "playing" with multiple rates, LRRL can accelerate progress when higher values are beneficial and switch to smaller ones when finer convergence is required. This adaptability allows it to escape poor local minima and achieve stronger convergence than any single fixed rate.

---

[1]AdamRel could be combined with decay; CLR could cycle over other ranges. We fix these choices to focus on the adaptation criterion.

- **Converging to the best candidate.** On functions where a single rate is clearly optimal (*e.g.*, Rosenbrock, Zakharov), LRRL adapts toward that rate, outperforming the second best performing candidate.

These experiments demonstrate that LRRL is not only robust to candidate quality but can also exploit multiple learning rates within a single run. Whereas a fixed learning rate must trade off between speed of convergence and stability, LRRL adapts dynamically, combining both advantages. Importantly, these results in stationary optimization confirm that LRRL's effectiveness is not limited to the non-stationary structure of RL, but reflects a more general capacity to adapt learning dynamics.

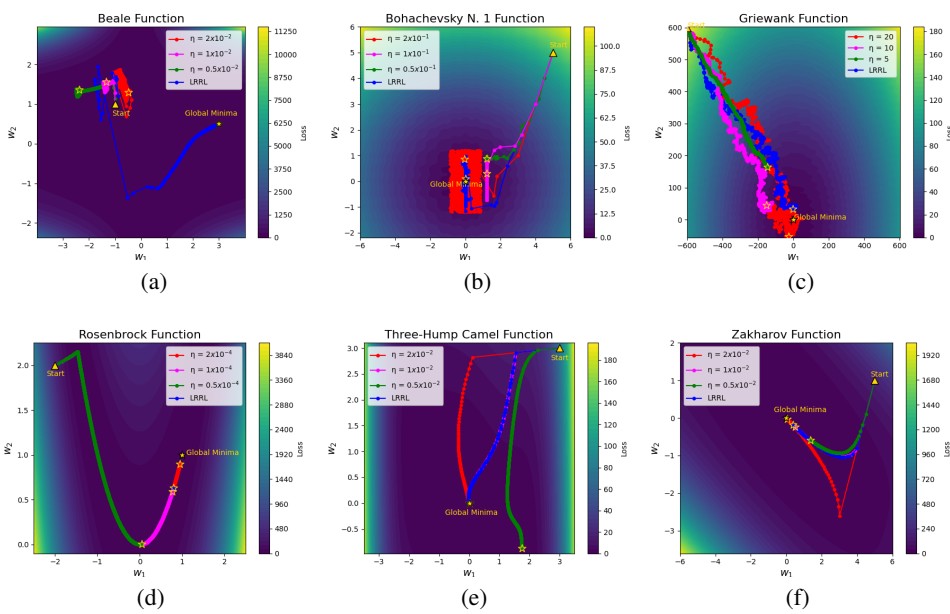

Figure 6: LRRL vs fixed learning rates in a stationary setting with SGD. LRRL with 3 rates is tested against the 3 constant rates for 6 non-convex losses (shown in Figure 11 in Appendix). LRRL is effective in either converging to the best performing rate or combining the set to achieve better results.

## 5 CONCLUSION

We introduced LRRL, a meta-learning approach that adapts the optimizer's learning rate on the fly using a multi-armed bandit. In contrast to fixed schedules or oracle tuning that require multiple runs, LRRL evaluates and selects among candidate rates within a single run, reducing tuning effort while remaining competitive or superior to the best fixed choice. Across discrete and continuous RL tasks, LRRL improved or matched strong baselines such as decay schedules, AdamRel, and CLR, and generalized to stationary non-convex optimization with SGD. These results highlight LRRL's ability to combine aggressive and conservative updates within a single trajectory, offering both robustness and adaptability.

**Limitations.** LRRL operates over a predefined set of candidate learning rates, which serves as the search space for adaptation. However, this requirement is inherent to any hyperparameter search, and our experiments show that LRRL can exploit multiple candidates dynamically, avoiding catastrophic failures and often outperforming fixed-rate baselines. Theoretical guarantees would require strong assumptions (e.g., convexity, smoothness, bounded losses) that deviate from deep RL; instead, we focused on broad empirical validation.

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

# A   RELATED WORK

This section presents works closely related to ours. We first introduce approaches that use MAB for hyperparameter selection, and later other methods that aim to adapt the learning rate over the training process.

**Multi-armed bandit for (hyper)parameter selection.**   Deep RL is known to be overly optimistic in the face of uncertainty (Ostrovski et al., 2021), and many works have addressed this issue by proposing conservative policy updates (van Hasselt et al., 2016; Fujimoto et al., 2019; Agarwal et al., 2020). However, when the agent can interact with the environment, this optimism might encourage exploration, potentially leading to the discovery of higher returns. Building on this idea, Moskovitz et al. (2021) proposes *Tactical Optimism and Pessimism* (TOP), an approach that uses an adversarial MAB algorithm to trade-off between pessimistic and optimistic policy updates based on the agent's performance over the learning process. Although our work shares similarities with TOP, such as employing an adversarial bandit algorithm and using performance improvement as feedback, there are key differences. First, TOP is closely tied to actor-critic methods, making it unclear how it could be extended to other RL paradigms. Moreover, it also introduces an exploration strategy which depends on epistemic uncertainty (captured through ensembles) and aleatoric uncertainty (estimated using a distributional RL algorithm). Second, TOP trades-off optimism and pessimism by adding new variables (arms), which are not inherently tied to the learning process and also require tuning. In contrast, LRRL offers an algorithm-agnostic solution by directly optimizing the learning rate, a core hyperparameter in all SGD-based methods, making it simpler and more broadly applicable. Also close to our work, Liu et al. (2020) propose Adam with Bandit Sampling (ADAMBS), which employs the Exp3 algorithm to enhance sample efficiency by incorporating importance sampling within the Adam optimizer. ADAMBS prioritizes informative samples by keeping a probability distribution over all samples, although the feedback is provided by the gradient computed over a mini-batch, unlike previous works in RL that are capable of assigning per-sample importance by using their computed TD-error (see Schaul et al., 2016).

In order to select the learning rate for stochastic gradient MCMC, Coullon et al. (2023) employs an algorithm based on Successive Halving (Karnin et al., 2013; Jamieson & Talwalkar, 2016), a MAB strategy that promotes promising arms and prunes suboptimal ones over time. In the context of hyperparameter optimization, Successive Halving has also been used in combination with infinite-arm bandits to select hyperparameters for supervised learning (Li et al., 2018; Shang et al., 2019). A key difference between these approaches to hyperparameter optimization for supervised learning and our work is that we focus on selecting the best learning rate on the fly from a predefined set, rather than performing an extensive and computationally expensive search over the hyperparameter space. Parker-Holder et al. (2020) propose population-based training which employs a Gaussian Process (see details in Rasmussen & Williams, 2005) with a time-varying kernel to represent the upper confidence bound in a bandit setting. The idea is to have multiple instances of agents learning a task in parallel, each with different hyperparameters, and replace those that are underperforming within a single run. Close to our work, MABSearch (Syed Shahul Hameed & Rajagopalan, 2024) frames the learning rate selection as a MAB. In practice, however, its algorithm reduces to an heuristic $\epsilon$-greedy scheme that maintains an exponential moving average of the loss to guide exploitation. As such, its performance depends critically on the choice of $\epsilon$ and its decay, lacking the principled, adaptive exploration–exploitation balance over the long run that defines proper bandit methods. In contrast, LRRL builds on adversarial algorithms such as Exp3, which maintain probabilistic weights over arms and update them under non-stationary feedback, providing the first rigorous integration of bandits as a meta-optimizer for learning rate adaptation.

**Learning rate adapters/schedulers.**   Optimizers such as RMSProp (Tieleman & Hinton, 2012) have an adaptive mechanism to update a set of parameters $\theta$ by normalizing past gradients, while Adam (Kingma & Ba, 2015) also incorporates momentum to smooth gradient steps. However, despite their widespread adoption, these algorithms have inherent limitations in non-stationary environments since they do not adapt to changes in the objective function over time (see Degris et al., 2024). Increment-Delta-Bar-Delta (IDBD), introduced by Sutton (1992), has an adaptive mechanism based on the loss to adjust the learning rate $\eta_i$ for each sample $x_i$ for linear regression and has been extended to settings including RL (Young et al., 2019). Learning rate schedulers with time decay (Senior et al., 2013; You et al., 2019) are coupled with optimizers, assuming gradual conver-

gence to a good solution, but often require task-specific manual tuning. A meta-gradient RL method is proposed in Xu et al. (2020), formulated as a two-level optimization process: one level optimizes the agent's policy objective, while the other learns meta-parameters of the objective function. Although both LRRL and meta-gradient approaches aim to enable agents to "learn how to learn" within a single agent's lifetime, meta-gradient methods must be tailored to the specific loss structure and update rules of each RL algorithm. In contrast, LRRL is algorithm-agnostic: it operates at the optimizer level, adapting the learning rate through a bandit mechanism without requiring gradients of the learning objective or modifications according to the underlying algorithm.

# B    SUPPLEMENTARY EXPERIMENTS

## B.1    BASELINE EVALUATION WITH VARYING LEARNING RATES

To establish baselines, we evaluate DQN with Adam across several Atari tasks using fixed learning rates. As shown in Figure 7, larger learning rates may provide a better jumpstart but often lead to worse asymptotic performance, while smaller rates result in more stable convergence but slower learning, with the best-performing learning rate varying by environment. This illustrates why a dynamic strategy such as LRRL, which is able to exploit higher rates early and smaller ones later, can be advantageous.

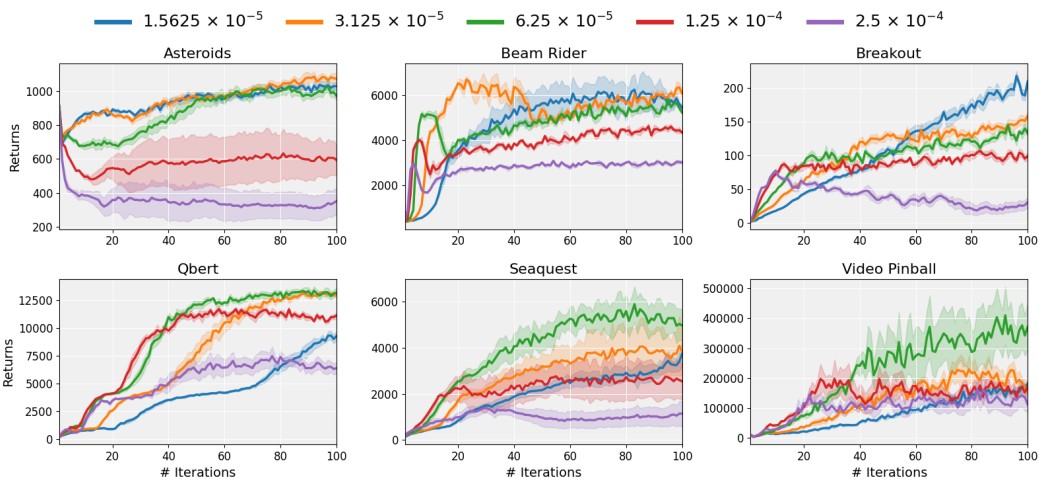

Figure 7: Baseline evaluation of DQN with Adam using different fixed learning rates across Atari tasks. Performance varies substantially across games, confirming that no single learning rate works universally.

## B.2    COMPARING ADVERSARIAL AND STOCHASTIC MAB ALGORITHMS

Adversarial MAB algorithms are designed for settings where reward distributions can change over time, making them naturally suited to RL. In contrast, stochastic MAB algorithms assume that rewards are drawn from fixed but unknown distributions. To validate our design choice and assess robustness, we compare LRRL using Exp3 with the stochastic MAB algorithm MOSS (Minimax Optimal Strategy in the Stochastic case) (Audibert & Bubeck, 2009).

MOSS balances exploration and exploitation by selecting the arm $k$ with the largest upper confidence bound:

$$B_k(n) = \hat{\mu}_k(n) + \rho \sqrt{\frac{\max\left(\log \frac{n}{Kn_k(n)}, 0\right)}{n_k(n)}}$$

where $\hat{\mu}_k(n)$ is the empirical average reward of arm $k$, and $n_k(n)$ is the number of times it has been pulled up to round $n$. In our experiments (Figure 8), we vary the step-size $\alpha$ for Exp3 and the exploration parameter $\rho$ for MOSS, using average cumulative return $f_n$ as feedback for MOSS.

Results show that Exp3 achieves stronger overall performance, consistent with the non-stationary nature of RL. Nevertheless, MOSS can reach competitive outcomes under appropriate exploration settings, demonstrating that LRRL is robust to the choice of bandit algorithm.

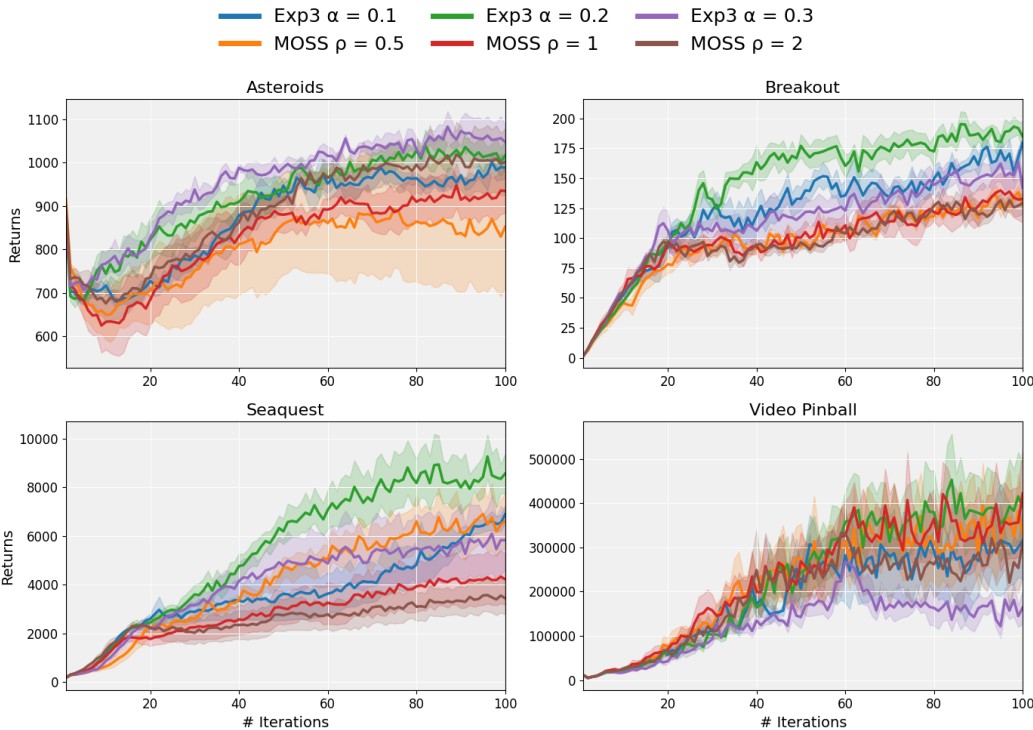

Figure 8: Comparison of adversarial (Exp3) and stochastic (MOSS) bandit algorithms within LRRL. Exp3 performs better overall, as expected in non-stationary RL settings, but MOSS can achieve competitive results under certain exploration parameters, highlighting robustness to the bandit choice.

## B.3 COMPARING RMSPROP AND ADAM OPTIMIZERS

We next evaluate LRRL's robustness to the choice of optimizer. We compare Adam and RMSProp augmented with Nesterov momentum (RMSProp-M), since prior work (Andrychowicz et al., 2021) and our own observations indicate that adding momentum consistently improves RMSProp's performance. Both Adam and RMSProp-M are paired with either LRRL and the best-performing fixed learning rate within the DQN agent. Since LRRL has already shown consistency in earlier sections, we extend the evaluation to three additional Atari games here.

As shown in Figure 9, LRRL combined with Adam achieves the strongest results overall, outperforming both RMSProp-M baselines and the tuned fixed-rate Adam baseline. LRRL with RMSProp-M provides better jumpstart but converges more slowly, underperforming standard DQN in two out of the three tasks. One possible explanation is the lack of bias correction in RMSProp's moment estimates, which may limit LRRL's ability to fully adapt. Exploring such optimizer-specific interactions is an interesting direction for future work.

## B.4 DISTRIBUTIONAL RL: IMPLICIT QUANTILE NETWORKS (IQN)

To further illustrate the generality of LRRL, we combine it with Implicit Quantile Networks (IQN) (Dabney et al., 2018), a distributional RL algorithm available in Dopamine that has shown

stronger empirical performance than DQN and other algorithms in several Atari tasks. Figure 10 shows that LRRL improves or matches performance across the evaluated Atari tasks when paired with IQN. These results demonstrate that LRRL is not tied to a specific algorithmic family: beyond standard value-based methods such as DQN or actor–critic methods such as PPO, LRRL can also enhance distributional agents. More broadly, since LRRL operates at the optimizer level by dynamically selecting the learning rate, it can in principle be coupled with any algorithm based on SGD or its variants. This makes LRRL widely applicable across deep RL.

### B.5    SUPPLEMENTARY RESULTS USING SGD

In Section 4.4, we tested LRRL with SGD on six non-convex loss landscapes (Figure 11). These benchmarks are widely used in optimization because their diverse shapes highlight different challenges, such as multiple local minima, steep ridges, or flat plateaus. Using SGD in this stationary setting minimizes the influence of confounders such as environment stochasticity in RL or the adaptivity of more sophisticated optimizers like Adam.

Here, we extend our experiments and compare LRRL against *Cyclical Learning Rates (CLR)* (Smith, 2017). CLR is motivated by a similar intuition that varying learning rates can improve performance, but it follows a predetermined periodic schedule that cycles through candidate values with a fixed step size $n$ (see Algorithm 2). Unlike LRRL, CLR does not use feedback to guide its choices. As a result, while CLR can occasionally reach strong solutions, it may introduce instability by switching rates at arbitrary times. In contrast, LRRL adapts its selection directly from performance feedback.

Figure 12 shows the gradient steps taken employing the best performing candidate, in terms of convergence rate, varying respectively the exploration trade-off given by $\rho$ (LRRL) and the step granularity $n$ (CLR). Table 1 displays the convergence steps to global minima considering loss

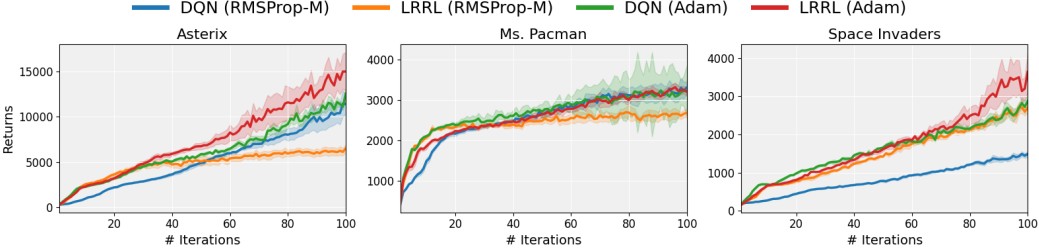

Figure 9: Comparison of Adam and RMSProp with Nesterov momentum (RMSProp-M), each combined with either LRRL or the best-performing fixed learning rate on DQN across three additional Atari games. LRRL with Adam consistently achieves the strongest results, while LRRL with RMSProp-M shows good jumpstart performance but slower convergence.

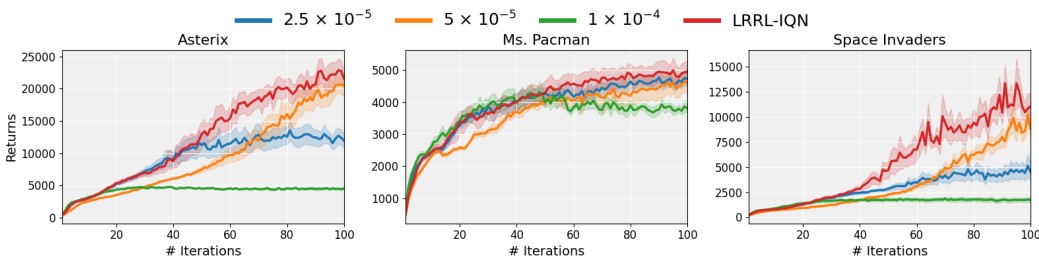

Figure 10: IQN with Adam on Atari tasks: LRRL vs. fixed learning rates. LRRL improves or matches performance across environments, showing that dynamic learning-rate adaptation benefits stronger distributional RL agents as well as DQN.

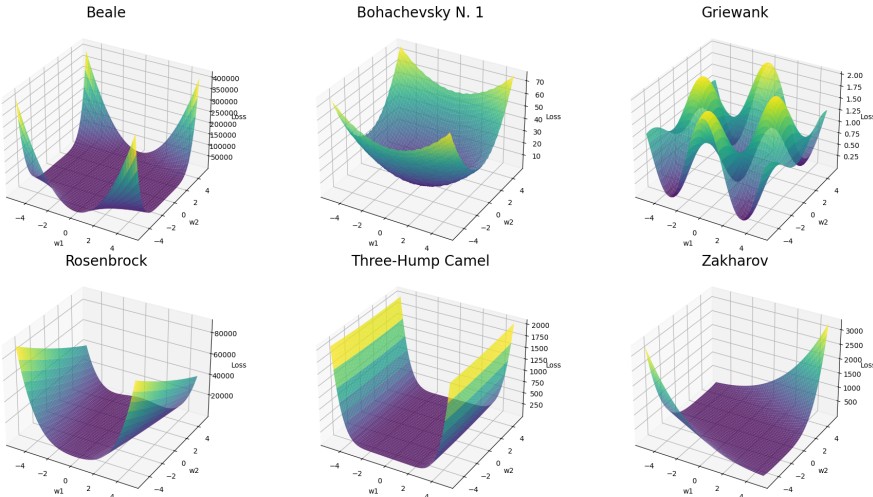

Figure 11: The six stationary non-convex loss landscapes used to assess LRRL with the SGD optimizer. Each function presents distinct challenges, such as narrow valleys (Rosenbrock) or multiple local minima (Griewank).

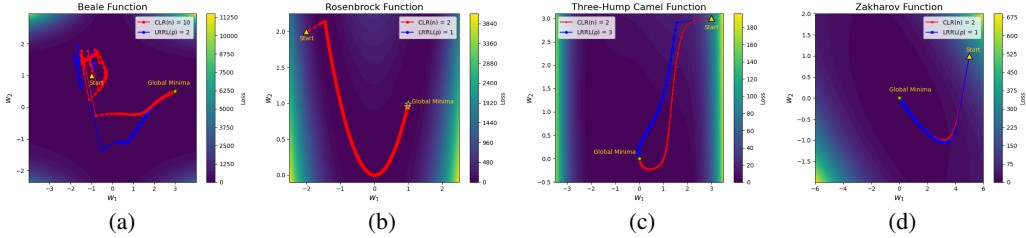

(a)       (b)       (c)       (d)

Figure 12: Convergence steps towards the global minima using either LRRL or CLR.

---

**Algorithm 2** Cyclical Learning Rates (Smith, 2017)

---

1: **Input:**
2:  $n$: step size (half-cycle length in iterations)
3:  minlr: minimum learning rate
4:  maxlr: maximum learning rate
5:  $M$: total number of training steps
6: **for** $m = 1, 2, \ldots, M$ **do**
7:  cycle $\leftarrow \left\lfloor 1 + \frac{m}{2n} \right\rfloor$   (Compute current half-cycle index)
8:  $x \leftarrow \left| \frac{m}{n} - 2 \cdot \text{cycle} + 1 \right|$   (Normalized position within cycle)
9:  $\eta \leftarrow \text{minlr} + (\text{maxlr} - \text{minlr}) \cdot \max(0, 1 - x)$   (Compute learning rate)
10: **end for**

---

landscapes where that could be achieved with an Euclidean distance less than $0.01$. Notably, LRRL has superior outcomes despite its higher sensitivity to the hyperparameter choice.

| Loss Landscape | CLR ($n = 2$) | CLR ($n = 5$) | CLR ($n = 10$) | LRRL ($\rho = 1$) | LRRL ($\rho = 2$) | LRRL ($\rho = 3$) |
|---|---|---|---|---|---|---|
| Beale | - | 1268 | 1248 | 2083 | **905** | 907 |
| Rosenbrock | 96267 | 96267 | 96275 | **65318** | 118167 | 66257 |
| Three-Hump Camel | 292 | 320 | 359 | 286 | 518 | **169** |
| Zakharov | 237 | 237 | 236 | **171** | 284 | 178 |

Table 1: Convergence steps to global minima (Euclidean distance $< 0.01$) for different loss landscapes and learning rate schedules.

## C  ADDITIONAL DISCUSSION AND LIMITATIONS

### C.1  ON SELECTING THE LEARNING RATES AND HYPERPARAMETERS

**Learning rates.**  Figure 7 shows that different learning rates lead to very different policy performance, yet there is no principled way to select them without extensive testing. LRRL does not eliminate the need to define a candidate set, and its performance naturally depends on this choice, as seen in Figure 1. However, LRRL reduces the burden of tuning by adaptively selecting among candidates within a single run, instead of requiring multiple runs to evaluate each rate individually. As illustrated in the SGD experiments (Figure 6), this allows LRRL to combine aggressive and conservative rates dynamically or to converge toward the best rate when one dominates.

**Hyperparameters.**  For the bandit algorithm, we fixed $\delta = 0.9$ in all experiments, a common choice for decay factors (*e.g.*, momentum or the RL discount factor $\gamma$). For the bandit feedback $f'_n$, a history window of $j = 5$ was robust across environments. As shown in Figure 4, $\kappa$ only slightly affects performance, while $\alpha$ is the most sensitive parameter (Figure 8), requiring tuning. This reflects a broader challenge in meta-optimization: hyperparameters are themselves required to control the process of tuning other hyperparameters (Mahmood et al., 2012).

### C.2  THE CUMULATIVE REWARD AS A PROXY OF PERFORMANCE

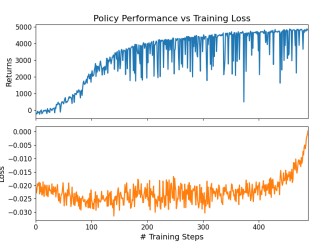

Figure 13: Return and loss in RL.

A central design choice in LRRL is the feedback used by the bandit algorithm. Our setting corresponds to a dual objective: the bandit aims to minimize regret in selecting learning rates, while the RL agent seeks to maximize cumulative returns. We therefore use the improvement in policy performance $f'_n$ as feedback to update arm probabilities.

Unlike supervised learning, where the training loss directly reflects predictive quality, in RL the optimization loss is only loosely related to policy performance (Figure 13). For example, the TD-error in Equation 1 measures how surprising a transition is, but a high TD-error does not necessarily indicate that the policy is improving. By contrast, returns directly measure policy effectiveness, even if they are noisy or sparse. This noise is intrinsic to RL and affects both LRRL's feedback and policy optimization itself.

### C.3  COMPUTATIONAL OVERHEAD

The overhead introduced by LRRL is governed primarily by the decision frequency $\kappa$ (how often we update arm probabilities) and the number of arms $K$. Each bandit update costs $O(K)$ for computing/sampling the distribution, so the total cost over $T$ decisions is $O(TK)$. In our SGD experiments, we update once per gradient step ($\kappa = 1$ step), so the extra $O(K)$ work can be noticeable when the model is tiny (*e.g.*, 2D landscapes) because the gradient step itself is very cheap. In RL, we set $\kappa = 1$ episode while policy updates happen every $\lambda = 4$ steps with mini-batches; thus $T$ is roughly the number of episodes. With small $K$ (as used here), the $O(TK)$ bandit cost is negligible compared to forward/backward passes and replay/batching, and adds no meaningful wall-clock overhead.

### C.4  GUARANTEES

While adversarial bandit algorithms offer theoretical regret guarantees under certain conditions, these guarantees do not carry over to deep RL because the best-performing arm itself changes as the policy evolves. From a theoretical point of view, guarantees would require assumptions (*e.g*, convexity, smoothness, bounded losses) that deviate significantly from the realities of RL. We therefore focus on empirical validation across diverse benchmarks, demonstrating LRRL's robustness and generality in practice.

## C.5 Future Work

A key limitation of LRRL is the need to predefine a set of candidate learning rates, although our experiments show some robustness to this choice. One possible direction to overcome this limitation is using Bayesian optimization, for example with Gaussian Processes (GPs) (Rasmussen & Williams, 2005). While GPs are sample-efficient, their computational cost increases as the number of observations grows, and they can struggle with non-stationary objectives. Addressing this would likely require mechanisms that emphasize recent interactions, as in time-varying GP-UCB (Bogunovic et al., 2016), or approaches that limit and/or adapt the number of inducing points (Pescador-Barrios et al., 2025). Integrating such techniques with LRRL could reduce reliance on a fixed candidate set while retaining adaptability to non-stationary dynamics.

## D Evaluation Metrics

In this section, we describe the evaluation metrics that can be used to evaluate the agent's performance as it interacts with an environment. Based on Dopamine, we use the evaluation step-size "iterations", defined as a predetermined number of episodes. In MuJoCo tasks, updates refer to policy updates. Figure 14 illustrates the evaluation metrics used in this work, as defined in Taylor & Stone (2009):

- **Max average return:** The highest average return obtained by an algorithm throughout the learning process. It is calculated by averaging the outcomes across multiple individual runs.

- **Final performance:** The performance of an algorithm after a predefined number of interactions. While two algorithms may reach the same final performance, they might require different amounts of data to do so. This metric captures the efficiency of an algorithm in reaching a certain level of performance within a limited number of interactions. In Figure 14, the final performance overlaps with the max average return, represented by the black dashed line.

- **Jumpstart performance:** The performance at the initial stages of training, starting from a policy with randomized parameters $\theta$. In Figure 14, Algorithm A exhibits better jumpstart performance but ultimately achieves lower final performance than Algorithm B. A lower jumpstart performance can result from factors such as a lower learning rate, although this work demonstrates that this does not necessarily lead to worse final performance.

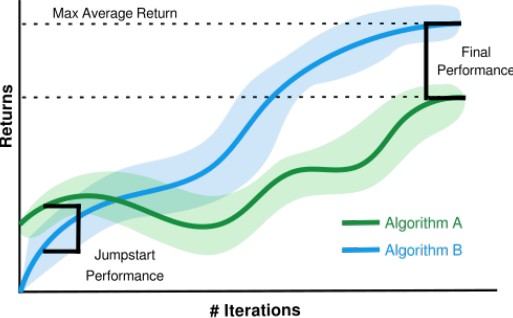

Figure 14: Performance curves of two RL algorithms (adapted from Taylor & Stone, 2009).

## E Implementation Details

In the following, we list the set of arms, optimizer and the bandit step-size used in each experiment.

**Section 4.1 – LRRL vs. Fixed Learning Rates: Robustness Across Candidate Sets**

- Optimizer: Adam
- Bandit step-size: $\alpha = 0.2$

- Considered sets of learning rates:

$$\mathcal{K}(5) = \left\{1.5625 \times 10^{-5}, 3.125 \times 10^{-5}, 6.25 \times 10^{-5}, 1.25 \times 10^{-4}, 2.5 \times 10^{-4}\right\}$$
$$\mathcal{K}_{\text{lowest}}(3) = \left\{1.5625 \times 10^{-5}, 3.125 \times 10^{-5}, 6.25 \times 10^{-5}\right\}$$
$$\mathcal{K}_{\text{middle}}(3) = \left\{3.125 \times 10^{-5}, 6.25 \times 10^{-5}, 1.25 \times 10^{-4}\right\}$$
$$\mathcal{K}_{\text{highest}}(3) = \left\{6.25 \times 10^{-5}, 1.25 \times 10^{-4}, 2.5 \times 10^{-4}\right\}$$
$$\mathcal{K}_{\text{sparse}}(3) = \left\{1.5625 \times 10^{-5}, \qquad 6.25 \times 10^{-5}, \qquad 2.5 \times 10^{-4}\right\}$$

**Section 4.2 – LRRL with Learning Rate Schedulers.**

- Optimizer: Adam
- Bandit step-size: $\alpha = 0.2$
- Initial learning rate of schedulers: $\eta_0 = 6.25 \times 10^{-5}$

**Section 4.3 – Continuous Action Spaces.**

- Optimizer: Adam
- Bandit step-size: $\alpha = 0.2$
- AdamRel/Initial learning rate: $3 \times 10^{-4}$
- LRRL/CLR set of learning rates: $\mathcal{K} = \left\{1.2 \times 10^{-4}, 1.8 \times 10^{-4}, 2.5 \times 10^{-4}\right\}$

**Section B.2 – Comparing adversarial and stochastic MAB algorithms.**

- Optimizer: Adam
- MOSS/Exp3 set of learning rates: $\mathcal{K} = \left\{3.125 \times 10^{-5}, 6.25 \times 10^{-5}, 1.25 \times 10^{-4}\right\}$

**Section B.3– Comparing RMSProp and Adam Optimizers.**

- Optimizer: RMSProp for DQN baselines
- Bandit step-size: $\alpha = 0.2$
- LRRL-DQN set of learning rates: $\mathcal{K} = \left\{1.5625 \times 10^{-5}, 3.125 \times 10^{-5}, 6.25 \times 10^{-5}\right\}$

**Section B.4– Distributional RL: Implicit Quantile Networks (IQN).**

- Optimizer: Adam
- Bandit step-size: $\alpha = 0.2$
- LRRL-IQN set of learning rates: $\mathcal{K} = \left\{2.5 \times 10^{-5}, 5 \times 10^{-5}, 1 \times 10^{-4}\right\}$

**Hyperparameters of the optimizers and deep RL algorithms.** For reproductibility, the list of hyperparameters of the optimizers and deep RL algorithms used for our experiments is provided in Table 2. The optimizers from the Optax library (DeepMind, 2020) are employed alongside the JAX (Bradbury et al., 2018) implementation of the DQN (Mnih et al., 2013; 2015) and IQN (Dabney et al., 2018) algorithms, as provided by the Dopamine framework (Castro et al., 2018). For the MuJoCo tasks, we use the implementation provided by TorchRL (Bou et al., 2024).

## F  USE OF LARGE LANGUAGE MODELS

In preparing this manuscript, we used large language model (LLMs) in the following ways:

1. **Text refinement.** The model was used to improve clarity, grammar, and flow, and to suggest alternative phrasings of drafts. It also helped organize reviewer-style feedback on earlier versions of the text.

| DQN and IQN hyperparameters | Value |
|---|---|
| Sticky actions | True |
| Sticky actions probability | 0.25 |
| Discount factor ($\gamma$) | 0.99 |
| Frames stacked | 4 |
| Mini-batch size ($\mathcal{B}$) | 32 |
| Replay memory start size | 20 000 |
| Learning network update rate ($\lambda$) | 4 steps |
| Minimum environment steps ($\kappa$) | 1 episode |
| Target network update rate ($\tau$) | 8000 steps |
| Initial exploration ($\epsilon$) | 1 |
| Exploration decay rate | 0.01 |
| Exploration decay period (steps) | 250 000 |
| Environment steps per iteration (steps) | 250 000 |
| Network neurons per layer | 32, 64, 64 |
| Hardware (GPU) | V100 |
| **Adam hyperparameters** | |
| $\beta_1$ decay | 0.9 |
| $\beta_2$ decay | 0.999 |
| Eps (DQN) | $1.5\times10^{-4}$ |
| Eps (IQN) | $3.125\times10^{-4}$ |
| **RMSProp hyperparameters (DQN)** | |
| Decay | 0.9 |
| Momentum (if *True*) | 0.999 |
| Centered | False |
| Eps | $1.5\times10^{-4}$ |

Table 2: Hyperparameters used in the experiments with discrete action spaces.

| PPO hyperparameters | Value |
|---|---|
| Discount factor ($\gamma$) | 0.99 |
| Mini-batch size ($\mathcal{B}$) | 64 |
| Epochs | 10 |
| GAE | 0.95 |
| Clip $\epsilon$ | 0.2 |
| Critic coef. | 0.25 |
| Entropy coef. | 0.0 |
| Minimum environment steps ($\kappa$) | 1 episode |
| Network neurons per layer | 64, 64 |
| Hardware (GPU) | V100 |

Table 3: Hyperparameters used in the experiments with continuous action spaces.

2. **Baseline verification.** The model was used to double check whether our implementations of CLR and AdamRel were aligned with their respective published descriptions.

3. **This disclosure.** To ensure transparency, this statement was written by the model, using the history of our interactions as context.

All research ideas, methodological contributions, experimental design, and analysis are our own. The LLM was not used to generate new technical content or experimental results. The authors take full responsibility for the content of this paper.

