# OpenReview forum: "Dynamic Learning Rate for Deep Reinforcement Learning: A Bandit Approach"
_ICLR.cc/2026/Conference — Submitted to ICLR 2026_

### Official Review · Reviewer_CaoU · 2025-10-21

**Soundness:** 2
**Presentation:** 2
**Contribution:** 2
**Rating:** 2
**Confidence:** 5

**Summary:**

This paper proposes LRRL, a bandit-based method, to dynamically adapt the learning rate for deep reinforcement learning.

The set of learning rates are predefined in advance by users. LRRL will consider each learning rate choice as an arm in the multi-arm bandit setting.

The experiments show that the LRRL can gain benefit against the fixed rate schedule.

The paper is generally well written and easy to follow.

However, the reviewer has a strong concern about the novelty, significant and lack of key baselines.

**Strengths:**

The idea is simple and effective

The experiment is better than the fixed rate schedule.

**Weaknesses:**

There are existing works, such as PB2 [1,2] have already solved the above problem in a much scalable and principled way.

* While this paper defines the fixed and discrete schedule for learning rate, PB2 [1] considers a continuous range [min, max] that allows the search operate flexibly. At the end, PB2 can return the optimal learning rate.

* While this paper only considers optimizing a single parameter, a learning rate, PB2 [1] can jointly optimize and learn the schedule for multiple parameters.

* The idea of using EXP3 algorithm as multi-arm bandit for selection has also been explored in PB2-Mix [2] in which the authors of PB2-Mix consider optimizing the mixed space of continuous and discrete variables.

* Both PB2 [1] and PB2-Mix [2] come with theoretical guarantee on the performance of the algorithm, while this paper LRRL does not have any guarantee.

* This paper doesnot compare with PB2.


[1] Parker-Holder, J., et al. (2020). Provably efficient online hyperparameter optimization with population-based bandits. NeurIPS 2020
[2] Parker-Holder, J , et al. "Tuning mixed input hyperparameters on the fly for efficient population based autorl." NeurIPS 2021

**Questions:**

Please see the weakness.
Consider a compare with PB2 and explain why LRRL is better than PB2?

Code from the RayTune library https://docs.ray.io/en/latest/tune/api/doc/ray.tune.schedulers.pb2.PB2.html
Code from the author https://github.com/jparkerholder/PB2

---

### Official Review · Reviewer_mGzu · 2025-10-28

**Soundness:** 2
**Presentation:** 2
**Contribution:** 2
**Rating:** 2
**Confidence:** 4

**Summary:**

This paper introduces LRRL, a bandit-based meta-learning approach that dynamically selects the learning rate during deep RL training based on policy performance feedback. Experiments are conducted using Atari, Mujoco, and stationary non-convex optimization problems, demonstrating promising performance compared to fixed learning rates and standard decay schedulers.

**Strengths:**

- In summary, this paper investigates the hyperparameter optimization (HPO) problem in deep RL. This is a promising area, and I really appreciate it. However, the whole paper focuses on optimizing the learning rate solely, which significantly restricts its landscape and scalability. Moreover, it seems that LRRL can only handle categorical HP values rather than continuous HP space.

- Bandit-based HPO approaches have been widely studied, such as Hyperband [1] and ULTHO [2]. Could the authors clarify the key advantage of LRRL compared to the previous work?

- Very few comparison experiment results are provided, which raises concerns about the superiority of LRRL. I recommend that the authors use ARLBench [3], a benchmark of HPO methods in RL, to conduct more comprehensive experiments.


[1] Li L, Jamieson K, DeSalvo G, et al. Hyperband: A novel bandit-based approach to hyperparameter optimization[J]. Journal of Machine Learning Research, 2018, 18(185): 1-52.

[2] Yuan M, Li B, Jin X, et al. ULTHO: Ultra-Lightweight yet Efficient Hyperparameter Optimization in Deep Reinforcement Learning[J]. arXiv preprint arXiv:2503.06101, 2025.

[3] Becktepe J, Dierkes J, Benjamins C, et al. ARLBench: Flexible and Efficient Benchmarking for Hyperparameter Optimization in Reinforcement Learning[J]. arXiv preprint arXiv:2409.18827, 2024.

**Weaknesses:**

See the comments above.

**Questions:**

See the comments above.

---

### Official Review · Reviewer_9a3E · 2025-10-31

**Soundness:** 2
**Presentation:** 2
**Contribution:** 2
**Rating:** 4
**Confidence:** 3

**Summary:**

This paper proposes using a multi-armed bandit algorithm to dynamically adjust an optimizer's learning rate in reinforcement learning. Concretely, the bandit selects from a finite set of learning rates (its 'arms') with the objective of maximizing "performance gains".

**Strengths:**

The proposed algorithm boosts the performance of a baseline RL algorithm without a significant computational overhead. Since the proposed algorithm is based on Exp3, an algorithm for a finite arm bandit, the computational overhead is significantly low compared to other meta-gradient methods.

**Weaknesses:**

- The goal is unclear.
- A heuristic definition of reward (Line 206) for learning rate tuning is not theoretically motivated well.
- Provided empirical results do not seem to be convincing.

# The Goal is Unclear

The research question asked in Introduction is the following: Can we adapt the learning rate dynamically based on the agent’s performance, rather than relying on training progress or gradient-based heuristics?

There would be many possible ways to adapt the learning rate dynamically based on the agent’s performance. Is the goal of the paper just finding one of them? I guess the authors actually want to find a better algorithm rather than just finding one of them. Then, a natural question is what metric to use. It seems the paper does not clearly mention which metric the paper cares about. Maybe regret?

Furthermore, the research question and empirical results do not seem to align well, if I do not misunderstand the paper. In Introduction, the authors wrote

> Because these methods rely on step counts or gradient norms rather than environment feedback, they often make premature or delayed adjustments, limiting their ability to cope with RL’s non-stationarity.

and asked the research question above. Therefore, I expected that the authors would have compared the combination of the proposed method with SGD (optimizer WITHOUT learning rate tuning based on training progress or gradient-based heuristics) to Adam/RMSProp (optimizer WITH learning rate tuning based on training progress or gradient-based heuristics). However, compared the combination of the proposed method with Adam/RMSProp to Adam/RMSProp.

# A Heuristic Definition of Reward

The reward definition (Line 206) for learning rate tuning is intuitive, but it is unclear whether it leads to any performance boost of a base RL algorithm. It would be nice if there exists any theory to justify it.

# Empirical Results not Convincing

Looking at figures, the performance gain by the proposed algorithm seems marginal.

In Figure 1, the authors might argue that their proposed method is on par with or superior to DQN with a BEST learning rate (Appendix B1). However, the fixed learning rate used there ($6.25 \times 10^{-5}$) is a default learning rate of DQN in Dopamine. Therefore, it is possible that DQN with Adam might stably work well with the learning rate $6.25 \times 10^{-5}$ across various environments. Then, we can simply use the learning rate $6.25 \times 10^{-5}$, and there is no need to tune the learning rate.

In Figure 3, the proposed method seem to clearly outperform other learning rate decays only in Breakout and Seaquest. Also in Breakout, DQN with a fixed learning rate (in Figure 1) seems to perform almost on par with the proposed algorithm. (Is the black dotted line calculated correctly?)

In Figure 5, the authors wrote that the proposed algorithm achieves higher returns on two of five MuJoCo task, but it is unclear if the proposed algorithm is statistically better.

**Questions:**

# Theoretical motivation for the choice of rewards for Exp3

Is it possible to theoretically motivate the choice of rewards for Exp3? For example, can we guarantee a higher return of a base RL algorithm under some ideal situation?

# Game where DQN with a default learning rate fails

Is there a game where DQN with a fixed learning rate of $6.25 \times 10^{-5}$ fails but LRRL performs better?

# Typo?

There seem to be some typos.

Firstly, in the definition of the pseudo-regret, I think $\max$ must be in front of $\sum$. See Section 11 of Lattimore & Szepesvari (2020) as well as Excercise 11.4 for an issue of using the regret defined as in the paper.

Secondly, in Equation 3, I think the denominator of $f'_n$ must be $p_n(k)$. The current one seems to result in a biased loss estimate.

---

### Official Review · Reviewer_owuz · 2025-11-02

**Soundness:** 2
**Presentation:** 2
**Contribution:** 2
**Rating:** 4
**Confidence:** 2

**Summary:**

* A modified version of Exp3 (the adversarial multi-armed bandit algorithm) for online adaptation (meta-learning) of the learning rate
* Tested on reinforcement learning (to maximize returns), and also on non-convex optimization problems

**Strengths:**

* The paper tests two codebases, two environment suites (Atari and MuJoCo), three base RL algorithms (DQN, IQN, and PPO), and three base optimizers (SGD, Adam, and RMSprop)
* Also tests six stationary, non-convex optimization problems
* To my limited knowledge, the approach is somewhat novel

**Weaknesses:**

* I think it overclaims the practical benefits. For example, "LRRL ... [reduces] tuning effort while remaining competitive or superior to the best fixed choice", but I think the empirical evidence for this is not that convincing. I think this overclaiming is a big weakness. For example, Fig. 8 shows the Exp3 learning rate is sensitive (which the paper itself notes).
* Few seeds (5-10)
* Adds 4 hyperparameters (alpha, delta, j, kappa)
* I am pretty sure "half of a standard deviation" is a nonstandard amount of variation, and I do not think it is justified in the paper
* The paper is framed around RL, but my understanding is the algorithm is not specialized to meta-learning the LR for RL, beyond the fact that it can handle nonstationarity and a non-differentiable final objective
* Compares against cyclical LR and AdamRel, but does not experimentally compare against other meta-learning algorithms for tuning the LR

**Questions:**

* "Similarly, adaptive optimizers such as RMSProp (Tieleman & Hinton, 2012) and Adam (Kingma & Ba, 2015) adjust effective learning rates (see Lyle et al., 2024)"
    * This confuses me. The "effective learning rate" in Lyle et al. is about neural networks that use normalization layers, which seems only tangentially related to adaptive optimizers. Lyle et al. do discuss the interaction of Adam with normalization layers, but still, the "effective learning rate" analysis there requires the normalization layers (or some similar, additional mechanism).
* "Because these methods rely on step counts or gradient norms rather than environment feedback, they often make premature or delayed adjustments, limiting their ability to cope with RL’s non-stationarity."
    * I think this should either be noted in the paper as speculation, or a reference should be provided for it
* "a dual objective"
    * I think the paper should avoid the term "dual" here, because "dual" often has a precise meaning in math that is not meant here
* "Unlike fixed schedules or gradient-based meta-learning heuristics, LRRL is algorithm-agnostic, seamlessly integrates with optimizers such as Adam, RMSProp, and SGD, and enables dynamic adaptation across different phases of training."
    * I'm unsure if I understand any of the three claims here, particularly for gradient-based meta-learning vs LRRL.
    * "algorithm-agnostic": what does this mean here? I could imagine some meanings where "algorithm-aware" is actually better than "algorithm-agnostic". Is it supposed to mean differentiable vs. not differentiable?
    * "seamlessly integrates with optimizers such as [...]": again, what does this mean here? PyTorch's autodiff can differentiate through all three of those optimizers.
    * "enables dynamic adaptation across different phases of training": how does gradient-based meta-learning not do this?
* "which deep RL mitigates by employing a large replay memory and calculating the target using a frozen network"
    * many deep RL algorithms do not use a replay memory nor a target network
* "MAB can be viewed as a special case of RL [...]"
    * "MAB is a special case of RL [...]"
* "single best arm, and [...] is the accumulated reward"
    * the inline equation there overlaps the line above it
* "updates —specifically"
    * typo? (nonstandard to have a space on only one side of a single em dash in a sentence)
* "updated recursively"
    * Would it not be equally valid (and possibly simpler and thus clearer) to just say, for example, "updated iteratively"?
* "providing a controlled setting where performance can be attributed solely to learning rate adaptation"
    * Although I think it's great to test on stationary non-convex optimization problems like this, I'm confused by this description. The paper might instead say "providing a simpler setting [...]"?
        * (To elaborate: in both these stationary optimization problems and the RL problems, the only thing the paper changes between the baseline and the proposed algorithm is the learning rate adaptation, right? Another way to put my confusion: if the paper also tested on, for example, stationary _convex_ problems, they would be even simpler settings than merely stationary _non-convex_ problems, and yet in all cases I would say "the performance can be attributed solely to learning rate adaptation".)
* "We consider five possible learning rates,"
    * To me, that subjectively feels like a very narrow range of learning rates. It's narrower than 1e-5 to 3e-4.
* Figure 6 looks interesting, but is hard to interpret, in part because there is so much visual overlap between trajectories. A table of AUC or final loss or something like that might help a lot.
* "(e.g., convexity, [...]"
    * This "e.g." is not italicized, whereas the other "e.g." instances (at least in the main text) are italicized

---

### Meta-Review · Area_Chair_13nJ · 2025-12-29

**Summary:**

The reviewers unanimously concluded that the manuscript is not currently suitable for publication.

**Reviewer Concerns:**

no concerns

**Reviewer Scores:**

not relevant

---

### Decision · Program_Chairs · 2026-01-26

Reject